# Sound-driven single-electron transfer in a circuit of coupled quantum rails

Shintaro Takada [1,2,8], Hermann Edlbauer [1,8], Hugo V. Lepage [3,8], Junliang Wang [1], Pierre-André Mortemousque[1], Giorgos Georgiou[1,4], Crispin H.W. Barnes[3], Christopher J.B. Ford [3], Mingyun Yuan[5], Paulo V. Santos[5], Xavier Waintal[6], Arne Ludwig[7], Andreas D. Wieck [7], Matias Urdampilleta[1], Tristan Meunier[1] & Christopher Bäuerle [1]*

Surface acoustic waves (SAWs) strongly modulate the shallow electric potential in piezo-electric materials. In semiconductor heterostructures such as GaAs/AlGaAs, SAWs can thus be employed to transfer individual electrons between distant quantum dots. This transfer mechanism makes SAW technologies a promising candidate to convey quantum information through a circuit of quantum logic gates. Here we present two essential building blocks of such a SAW-driven quantum circuit. First, we implement a directional coupler allowing to partition a flying electron arbitrarily into two paths of transportation. Second, we demonstrate a triggered single-electron source enabling synchronisation of the SAW-driven sending process. Exceeding a single-shot transfer efficiency of 99%, we show that a SAW-driven integrated circuit is feasible with single electrons on a large scale. Our results pave the way to perform quantum logic operations with flying electron qubits.

[1] Université Grenoble Alpes, CNRS, Institut Néel, 38000 Grenoble, France. [2] National Institute of Advanced Industrial Science and Technology (AIST), National Metrology Institute of Japan (NMIJ), 1-1-1 Umezono, Tsukuba, Ibaraki 305-8563, Japan. [3] Cavendish Laboratory, Department of Physics, University of Cambridge, Cambridge CB3 0HE, UK. [4] Université Savoie Mont-Blanc, CNRS, IMEP-LAHC, 73370 Le Bourget du Lac, France. [5] Paul-Drude-Institut für Festkörperelektronik, Hausvogteiplatz 5-7, 10117 Berlin, Germany. [6] Université Grenoble Alpes, CEA, IRIG-Pheliqs, 38000 Grenoble, France. [7] Lehrstuhl für Angewandte Festkörperphysik, Ruhr-Universität Bochum, Universitätsstraße 150, 44780 Bochum, Germany. [8]These authors contributed equally to this work: Shintaro Takada, Hermann Edlbauer, Hugo V. Lepage. *email: christopher.bauerle@neel.cnrs.fr

iVincenzo's criteria for realising a quantum computer address the transmission of quantum information between stationary nodes[1]. Several approaches have demonstrated successful transmission of quantum states in solid-state devices such as in quantum dot (QD) arrays[2–5], coupled QDs in quantum Hall edge channels[6] or microwave-coupled superconducting qubits[7,8]. In semiconductor heterostructures, surface acoustic waves (SAWs) offer a particularly interesting platform to transmit quantum information. Thanks to the shallow electric potential modulation on a piezoelectric substrate, a SAW forms a train of moving QDs along a depleted transport channel. This SAW train allows to drag single charge carriers from one side of such a quantum rail to the other. Employing stationary QDs as electron source and receiver, a single electron has been sent back and forth several micrometre long tracks with a transfer efficiency of about 92%[9,10]. Recently, SAW-driven transfer of individual spin polarised electrons has been reported[11]. These advances support the idea of a SAW-driven quantum circuit enabling the implementation of electron-quantum-optics experiments[12–14] and quantum computation schemes at the single-particle level[15–19].

The core of such a quantum circuit is a tunable beam-splitter permitting the coherent partitioning and coupling of single-flying electrons. In the past, coherent quantum phenomena such as the Hanbury-Brown–Twiss or the Hong–Ou–Mandel effect have been observed by analysing fluctuations in current through a beam-splitter structure[20,21]. Inspired by these experiments, a refined beam-splitter geometry has been developed to demonstrate the basic principles of flying charge qubit manipulations in a Mach–Zehnder interferometry set-up with a continuous stream of ballistic electrons[22,23]. This progress moreover opened up the way for precise transmission-phase measurements of QD states[24–26] and detailed studies on quantum phenomena such as the Kondo effect[27,28]. Considering the coherence times in stationary charge[29–32] or spin qubits[33–35], it should be possible to use a surface-gate-defined beam-splitter component to implement quantum logic gates in GaAs-based heterostructures for solitary flying electron qubits. First steps in this directions have already been achieved via the demonstration of electron-quantum-optics experiments such as Hong–Ou–Mandel interference[12,36] or quantum state tomography[37–39]. To perform quantum logic operations[40] with a solitary flying electron qubit that is defined via charge or spin, besides coherent propagation of the electron wave function and single-shot detection, it will be further necessary to establish an experimental frame allowing adiabatic transport of the respective two-level system. Owing to the electrostatic isolation from the Fermi sea, SAW-driven single-electron transport is promising to demonstrate quantum logic operations with a flying electron qubit in a beam-splitter set-up.

In this work we investigate the feasibility of such a beam-splitter set-up for SAW-driven single-shot transfer of a solitary electron. For this purpose, we couple a pair of quantum rails by a tunnel-barrier and partition an electron in flight into the two output channels of the circuit. Modelling the experimental results of this directional-coupler operation with quantum mechanical simulations, we deliver insight into the quantum state of the SAW-transported electron and provide a clear route to maintain adiabatic transport along a tunnel-coupled region of quantum rails. In order to realise quantum logic gates, where a pair of electrons is made to interact in flight, it is further necessary to synchronise the sending process. For this purpose, we demonstrate a SAW-driven single-electron source that is triggered by a voltage pulse on a timescale of picoseconds.

## Results

**A sound-driven single-electron circuit.** The sample is realised via surface electrodes forming a depleted potential landscape in the two-dimensional electron gas (2DEG) of a GaAs/AlGaAs heterostructure. An interdigital transducer (IDT) is used to send a finite SAW train towards our single-electron circuit as shown schematically in Fig. 1a. A scanning-electron-microscopy (SEM) image of the investigated single-electron circuit is shown in Fig. 1b. The device consists of two 22-μm-long quantum rails that are coupled along a region of 2 μm by a tunnel-barrier, which is defined by a 20 nm -wide surface gate. The SAW train allows the transport of a single electron from one gate-defined QD (source) to another stationary QD (receiver) through the circuit of coupled quantum rails (QR). Figure 1c shows a zoom on the lower receiver QD with indications of the electrical connections. To detect the presence of an electron, a quantum point contact (QPC) is placed next to each QD. By biasing this QPC at a sensitive working point, an electron leaving or entering the QD can be detected by a jump in the current $I_{QPC}$[41].

**Transfer efficiency.** Let us first quantify the efficiency of SAW-driven single-electron transfer along a single quantum rail. For this purpose, we decouple the two transport channels by setting a high tunnel-barrier potential using a gate voltage of $V_T = -1.2$ V. To quantify the errors of loading, sending and catching, we repeat each SAW-driven transfer sequence with a reference experiment where we initially do not load an electron at the source QD. Figure 1d shows the jump in QPC current, $\Delta I_{QPC}$, after SAW transmission at the upper receiver QD for an exemplary set of thousand single-electron transfer experiments in an optimised configuration. The grey data points stem from the reference experiments without initial loading at the source QD. The distinct peaks in the histograms of the events with (red) and without (grey) initial loading show that the presence of an electron in the QD is clearly distinguishable. Analysing 70,000 successive experiments of this kind in a single optimised configuration of the quantum rail, we quantify the efficiency of SAW-driven single-electron transport. Thanks to the low error rates of loading (0.07%) and catching (0.18%), we deduce a transfer efficiency along our 20 μm-long quantum rail of 99.75%. A similar single-shot transfer efficiency has recently been obtained with single-electron pumps emitting high-energy ballistic electrons[42].

**Partitioning an electron in flight.** Having established highly efficient single-electron transport, we now couple the two channels to partition an electron in flight between the two quantum rails. The aim of this directional coupling is to prepare a superposition state of a flying electron qubit. We find that we can finely control the partitioning of the electron by detuning the double-well potential as indicated in Fig. 2a, b. To achieve this effect, we sweep the voltages applied to the side electrodes of the coupling region, $V_U$ and $V_L$, in opposite directions while keeping $V_T$ constant. With a potential detuning, $\Delta = V_U - V_L = 0$ V, the quantum rails are aligned in electric potential. Setting a voltage configuration where $\Delta < 0$, the potential of the lower quantum rail (L) is decreased with respect to the upper path (U). For $\Delta > 0$, the situation is reversed. Deducing the transfer probabilities to the receiver QDs from a thousand single-shot experiments per data point, we measure the partitioning of the electrons for different values of $\Delta$ as shown in Fig. 2c. Here, we sweep $V_U$ and $V_L$ in opposite directions from $-1.26$ V to $-0.96$ V while keeping $V_T = -0.75$ V. The data shows a gradual transition of the electron transfer probability from the upper (U) to the lower (L)

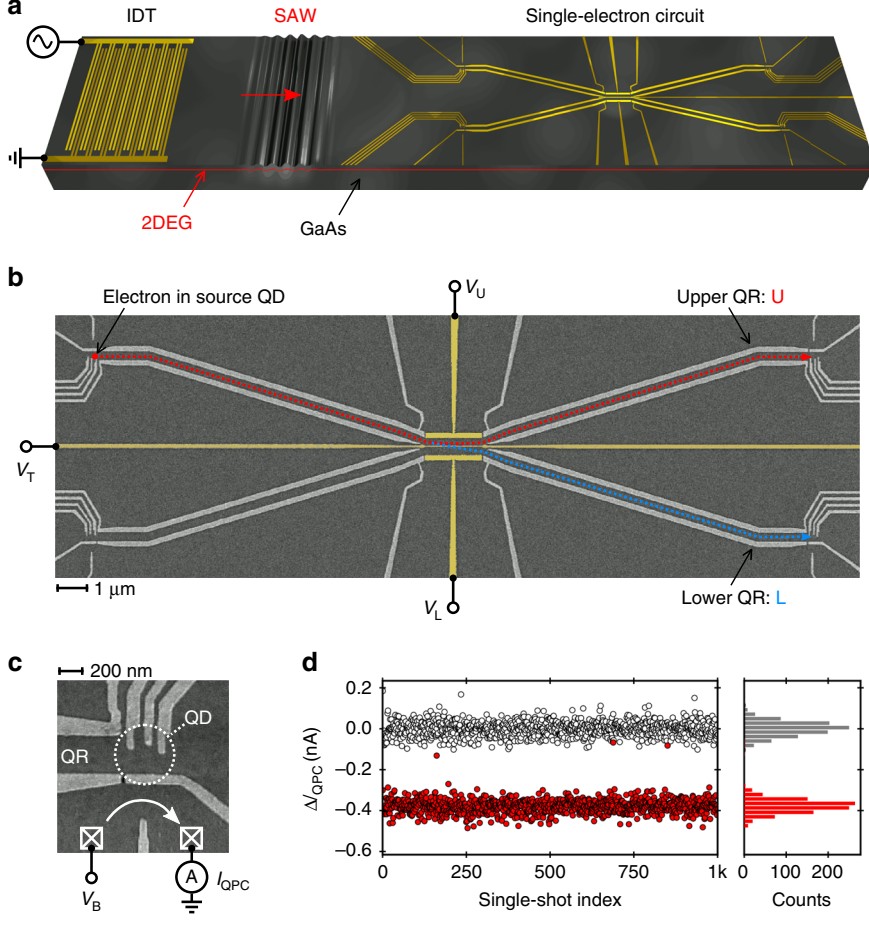

**Fig. 1** Sound-driven circuit of coupled quantum rails. **a** Schematic of the experimental set-up. An interdigital transducer (IDT) launches a SAW train towards the single-electron circuit, which is realised via metallic surface gates in a GaAs/AlGaAs heterostructure. **b** SEM image of the quantum rails (QR) with indications of the transport paths, U and L, and the voltages to control the coupling region. **c** SEM image of the lower receiver quantum dot (QD) with indication of the coupled quantum rail (QR) and the close-by quantum point contact (QPC). **d** Jumps in QPC current, $I_{QPC}$, at the upper receiver QD from thousand SAW-driven single-shot transfers with (red) and without (grey) initial loading of a solitary electron at the source QD

detector QD while the total transfer efficiency stays at $99.5 \pm 0.3\%$.

An interesting feature of the observed probability transition is that it follows the course of a Fermi–Dirac distribution:

$$P_{\mathrm{U}}(\Delta) \approx \frac{1}{\exp(-\Delta/\sigma) + 1} \quad (1)$$

Fitting the experimental data with such a function (see lines in Fig. 2c), we can quantify the width of the probability transition via the scale parameter, $\sigma$. To test the dependencies of the directional-coupler transition on the different properties of the device, we experimentally investigated if the width of the probability transition changes as we sweep the gate voltage configurations on different surface electrodes of the nanostructure. We find a significant narrowing of the probability transition (see Fig. 2d) as we increase the tunnel-barrier potential.

**The role of excitation**. To obtain a better understanding of our experimental observations, we first investigate the partitioning process by means of a stationary model. We consider a one-dimensional cut of the double-well potential in the tunnel-coupling region. In this region, we have a sufficiently flat potential landscape, $U(\boldsymbol{r}, t) \approx U(y) + U_{\mathrm{SAW}}(x, t)$, such that the eigenstate problem becomes separable in the $x$ and $y$ coordinates. The electronic wave function $\phi_i(y)$ along the transverse $y$ direction

satisfies the one-dimensional Schrödinger equation:

$$\frac{\hbar^2}{2m^*} \frac{\partial^2 \phi_i(y)}{\partial y^2} + U(y) \cdot \phi_i(y) = E_i \phi_i(y) \quad (2)$$

where $U(y)$ is a the electrostatic double-well potential for a given set of surface-gate voltages $V_{\mathrm{U}}$, $V_{\mathrm{L}}$ and $V_{\mathrm{T}}$. $m^*$ indicates the effective electron mass in a GaAs crystal. Here, we obtain $U(y)$ for the specific geometry of the presently investigated device by solving the corresponding Poisson problem[43,44].

To obtain the probability of finding the electron in the upper or lower potential well, we can now simply sum up the contributions of the wave function in the eigenstates for the respective region of interest. For the upper quantum rail, we integrate the modulus squared of the wave function over the spatial region of the upper quantum rail:

$$P_{\mathrm{U}} = \sum_i p_i \int\limits_{y > 0 \text{ nm}} |\phi_i(y, U(y))|^2 \, \mathrm{d}y \quad (3)$$

where $p_i$ is the occupation of the eigenstate $\phi_i$. For a fixed tunnel-barrier height, we can detune the double-well potential by varying $\Delta$, as in experiments. It is now straightforward to calculate the directional-coupler transition for the experimental setting with any imaginable occupation of the eigenstates.

Let us first consider the hypothetical situation where only the ground state is occupied. We evaluate Eq. (3) with mere ground

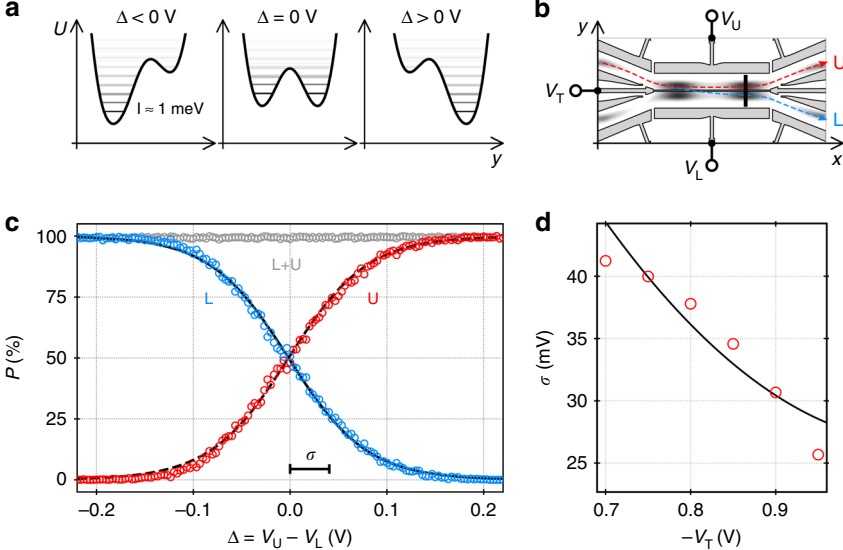

**Fig. 2** Directional-coupler operation. **a** Schematic slices along the double-well potential, $U$. The horizontal lines represent the eigenstates in the moving QD, whereas the grey shading of the energy levels indicates the exponentially decreasing occupation. **b** Schematic showing the QDs that are formed by the SAW in the coupling region with additional indications of the surface gates and the transport paths. The black vertical bar indicates the positions of the aforementioned potential slices. **c** Probability, $P$, to end up in the upper (U) or lower (L) quantum rail for different values of potential detuning, $\Delta$. The lines show a fit by a Fermi function providing the scale parameter, $\sigma$. **d** Transition widths, $\sigma$, for different values of the tunnel-barrier voltage, $V_T$. The line shows the course of a stationary, one-dimensional model of the partitioning process

state occupation ($p_0 = 1$) and fixed barrier potential ($V_T = -0.7$ V) for different values of potential detuning, $\Delta$, that are changed as in experiment. Doing so, we obtain a course of the probability transition having the shape of the aforementioned Fermi–Dirac distribution. Assuming ground state occupation in the double-well potential, we obtain however an extremely abrupt transition in transfer probability with a width, $\sigma$, that is in the order of several microvolts what is much smaller than in our experiment.

Let us now investigate how the situation changes as we populate successively excited eigenstates of the double-well potential. For this purpose we define the occupation of the eigenstates, $\phi_i$, with eigenenergies, $E_i$, via an exponential distribution:

$$p_i \propto \exp\left(-\frac{E_i - E_0}{\varepsilon}\right) \qquad (4)$$

where $\varepsilon$ is a parameter determining the occupation of higher energy eigenstates. This approach allows us to maintain the course of a Fermi distribution as we successively occupy excited states. Increasing the occupation parameter $\varepsilon$, we find a broadening of the probability transition. For $\varepsilon = 3.5$ meV we obtain simulation results showing very good agreement with the experimental data. Keeping $\varepsilon$ constant, the one-dimensional model follows the experimentally observed transition width, $\sigma$, over a wide range of $V_T$ as shown by the line in Fig. 2d. Note, however, that $\varepsilon$ only provides a rough estimate for the excitation energy that is present in our experiment due to the uncertainties that enter the model via the potential calculation. The model shows that the width of the directional-coupler transition, $\sigma$, reflects the occupation of excited states and thus indirectly the confinement in the moving QDs that are formed by the SAW along the tunnel-coupled quantum rails.

Our analysis of the experimental data shows that the flying electron is significantly excited as it propagates through the coupling region of the present circuit. To find possible sources of charge excitation, we employed a more elaborate model to simulate the time-dependent SAW-driven propagation of the

electron along different sections of our beam-splitter device[45]. For this purpose, we superimpose the static, two-dimensional potential landscape, $U(\mathbf{r})$, with the dynamic modulation of a SAW train, $U_{SAW}(x, t)$, that we estimate from Coulomb-blockade measurements. Simulating the entrance of a flying electron from the injection channel into the tunnel-coupled region, we find significant excitation of the flying electron into higher energy states.

To quantify adiabatic transport of the flying charge qubit, we define the qubit fidelity, $F$, as projection of the electron wave function on the two lowest eigenstates of the moving QD potential that is formed by the SAW along the coupled quantum rails. Figure 3a shows courses of the qubit fidelity, $F$, of a flying electron state that propagates along the tunnel-coupled region for different values of peak-to-peak SAW amplitude, $A$. For the present experiment, we estimate $A$ as 17 meV. For this value (red solid line), the simulation data shows an abrupt reduction of the qubit fidelity, $F$, due to the aforementioned excitation of the SAW-transported electron at injection from a single-quantum rail into the tunnel-coupled region. In congruence with the stationary, one-dimensional model that we applied before, the coupling into higher energy states leads to a spreading over both sides of the double-well potential as shown in Fig. 3b and Supplementary Movie 1. The simulation thus shows up a major source of excitation. When the electron passes from the strongly confined injection channel into the wide double-well potential it experiences an abrupt reconfiguration of the eigenstates in the moving QD what causes Landau–Zener transitions in higher energy states.

**Towards adiabatic transport**. Let us now investigate if we can reduce the probability for charge excitation by increasing the longitudinal confinement via the SAW amplitude. For $A = 30$ meV—see red dashed line in Fig. 3a—charge excitation is already strongly mitigated. The qubit fidelity vanishes however also in this case, since the electron still occupies low-energy states above the two-level system we are striving for. Despite non-adiabatic transport, we can already recognise coherent tunnel

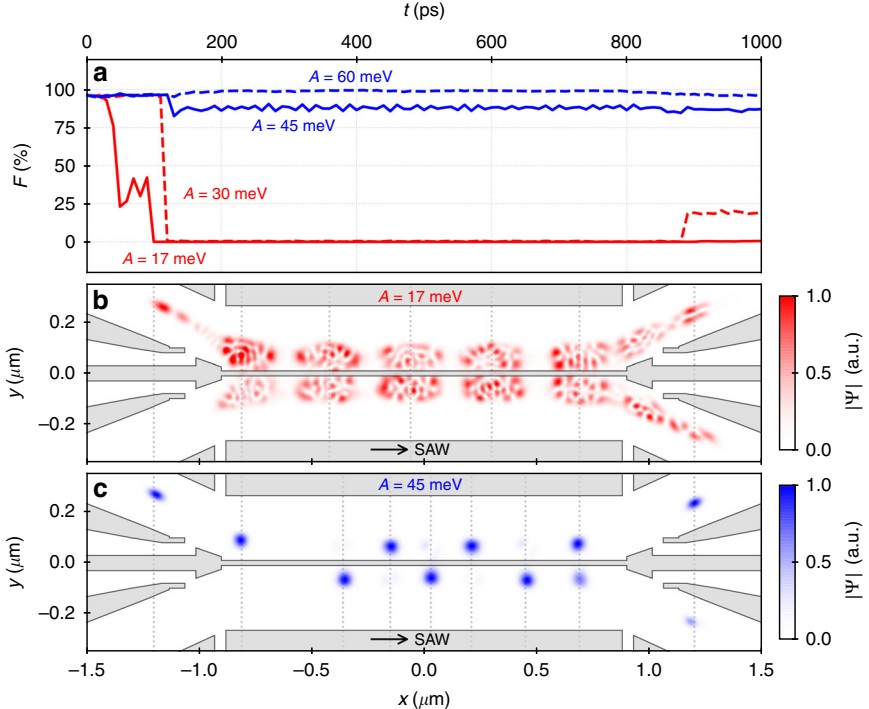

**Fig. 3** Time-dependent simulation of electron propagation. **a** Course of the qubit fidelity, $F$, for SAW-driven single-electron transport along the coupling region for different values of SAW amplitude, $A$. **b** Trace of the electron wave function, $\Psi$, along the coupled quantum rails for $A = 17$ meV at selected times, $t$, indicated via the vertical dashed lines. The grey regions indicate the surface gates. **c** Trace of $\Psi$ for $A = 45$ meV

oscillations when looking at the trace of the wave function as shown in Supplementary Movie 2. This shows that also excited electron states can undergo coherent tunnelling processes as previously expected in magnetic-field-assisted experiments on continuous SAW-driven single-electron transport through a quantum rail that is tunnel-coupled to an electron reservoir[46]. Increasing the SAW amplitude further to $A = 45$ meV (blue solid line), the transport of the electron gets nearly adiabatic and clear coherent tunnel oscillations occur as shown in Fig. 3c and Supplementary Movie 3. The simulations show that stronger SAW confinement can indeed prohibit charge excitation and maintain adiabatic transport. In experiment, one can increase the SAW confinement via many ways such as reduced attenuation of the IDT signal, longer IDT geometries, impedance matching or the implementation of more advanced SAW generation approaches[47–49]. We anticipate therefore the experimental observation of coherent tunnel oscillations in follow-up investigations.

**Triggering single-electron transfer**. Achieving adiabatic single-electron transport, a SAW train could also be employed to couple a pair of flying electrons in a beam-splitter set-up. In the long run, this coupling could enable entanglement of single-flying electron qubits through their Coulomb interaction[14] or spin[15]. For this purpose, electrons must be sent simultaneously from different sources in a specific position of the SAW train. Let us now investigate if we can achieve such synchronisation by using a fast voltage pulse as trigger for the sending process with the SAW[9]. After loading an electron from the reservoir, we bring the particle into a protected configuration where it cannot be picked up by the SAW. To load the electron into a specific minimum of the SAW train, we then apply a voltage pulse at the right moment to the plunger gate of the QD as schematically indicated in Fig. 4a, b. This pulse allows the electron to escape the stationary source QD into a specific moving QD formed by the SAW along the quantum rail.

To demonstrate the functioning of this trigger, we use a very short voltage pulse of 90 ps corresponding to a quarter SAW period[50]. Sweeping the delay of this pulse, $\tau$, over the arrival window of the SAW at the source QD, we observe distinct fringes of transfer probability as shown in Fig. 4c and more detailed in Fig. 4d. The data shows that the fringes are exactly spaced by the SAW period. The periodicity of the transmission peaks indicates that there is a particular phase along the SAW train where a picosecond pulse can efficiently transfer an electron from the stationary source QD into a specific SAW minimum. As the voltage pulse overlaps in time with this phase, the sending process is activated and the transfer probability rapidly goes up from $2.7 \pm 0.5\%$ to $99.0 \pm 0.4\%$. The finite background transfer probability is due to limited pulse amplitude in the present set-up. The envelope of the transfer fringes is consistent with the expected SAW profile. Comparing the directional-coupler measurement with and without triggering of the sending process, we find no change in the transition width what indicates that excitation at the source QD is comparably small or not present. By reduction of pulse attenuation along the transmission lines and optimisation of the QD structure, we anticipate further enhancements in the efficiency of the voltage-pulse trigger. The present pulsing approach allows us to synchronise the SAW-driven sending process along parallel quantum rails and represents thus an important milestone towards the coupling of single-flying electrons.

## Discussion

A flying qubit architecture is an appealing idea to transfer and manipulate quantum information between stationary nodes of computation[1,14,16]. Thanks to the isolation during transport and the availability of highly efficient single-electron sources and receivers, SAWs represent a particularly promising candidate to deliver the first quantum logic gate for electronic flying qubits[14,22,23]. Here, we have presented important milestones to

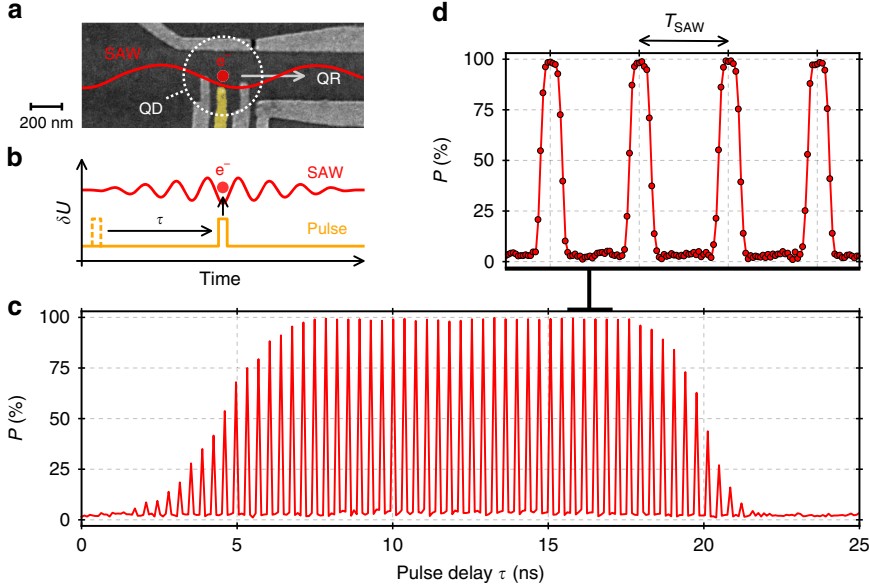

**Fig. 4** Pulse-triggered single-electron transfer. **a** SEM image of the source quantum dot (QD) showing the pulsing gate highlighted in yellow. A fast voltage pulse on this gate allows one to trigger SAW-driven single-electron transport along the quantum rail (QR) as schematically indicated. **b** Measurement scheme showing the modulation, $\delta U$, of the electric potential at the stationary source QD: the delay of a fast voltage pulse, $\tau$, is swept along the arrival window of the SAW. **c** Measurement of the probability, $P$, to transfer a single electron with the SAW from the source to the receiver QD for different values of $\tau$. **d** Zoom in on a time frame of four SAW periods, $T_{SAW}$

achieve this goal. First, we demonstrated the capability of the present device to partition a single-electron arbitrarily from one quantum rail into the other while maintaining a transfer efficiency above 99%. Employing quantum mechanical simulations, we reproduced the experimentally observed directional-coupler transition and identified charge excitation as remaining challenge for adiabatic transport through the coupling region of a SAW-driven single-electron circuit. Simulating SAW-driven electron propagation through the coupling region, we identified the central source of excitation and provided a clear route to remedy this problem in future investigations. We anticipate that an optimised surface-gate geometry as well as stronger SAW confinement[47–49] will allow coherent manipulation of a single electron in a true two-level state[29–32]. We demonstrated furthermore a powerful tool to synchronise the SAW-driven sending process along parallel quantum rails using a voltage-pulse trigger. With this achievement, we fulfil an important requirement to couple a pair of single electrons in a beam-splitter set-up. Our results pave the way for electron-quantum optics experiments[14] and quantum logic gates with flying electron qubits[40] at the single-particle level.

## Methods

**Experimental set-up**. The experiments are performed at a temperature of about 10 mK using a $^3$He/$^4$He dilution refrigerator. The present device is realised by a Schottky gate technique in a two-dimensional electron gas (2DEG) of a GaAs/AlGaAs heterostructure. The 2DEG is located at the GaAs/AlGaAs interface 100 nm below the surface and has an electron density of $n \approx 2.7 \times 10^{11}$ cm$^{-2}$ and a mobility of $\mu \approx 10^6$ cm$^2$ V$^{-1}$s$^{-1}$. It is formed by a Si-$\delta$-doped layer that is located 55 nm below the surface. All nanostructures are realised by Ti/Au electrodes (Ti: 5 nm; Au: 20 nm) that are written by electron-beam lithography on the surface of the wafer. Applying a set of negative voltages on these surface electrodes, we deplete the underlying 2DEG and form the potential landscape defining our beam-splitter device. Along the quantum rails there are thus no electrons present. The SAW-transported electron is thus completely decoupled from the Fermi sea.

The interdigital transducer (IDT) that we employ as source of a SAW train is placed outside of the mesa—about 1.6 mm beside the single-electron circuit. It contains 120 interdigitated double fingers with a finger spacing and width of 125 nm. The wavelength of the generated SAW is thus 1 μm. The aperture of the IDT fingers is 50 μm. We operate the device with a pulse-modulated, sinusoidal voltage signal oscillating at the IDT's resonance frequency of 2.77 GHz. In all of the present experiments, the duration of each oscillation pulse on the IDT was set

to 30 ns. The power on the signal generator was set to 25 dBm. We attenuate the IDT signal along the transmission line at two temperature stages in total by 8 dB to mitigate the injection of thermal noise. The propagation of evanescent electromagnetic waves from the IDT is suppressed by grounded metal shields. The jitter of the voltage pulse that we send from an arbitrary-waveform-generator (AWG) to the plunger gate of the source QD was measured as about 6.6 ps (FWHM) with respect to a fixed phase of the SAW burst.

**SAW-driven single-electron transfer**. To execute the sound-driven transport of a single electron, we perform a sequence of voltage movements on the surface gates defining the source and receiver QDs. In each single-shot-transfer experiment, we perform three steps before launching the SAW train: initialisation, loading and preparation to send. These steps are executed by fast voltage changes on the QD gates R and C as indicated in the SEM image shown in Fig. 5a. In between each step we go to a protected measurement configuration (M) and read out the current through the quantum point contact (QPC) as indicated in the charge-stability diagram shown in Fig. 5b. Comparing the QPC current before and after each step, we can deduce if an electron entered or left the QD.

To initialise the system, we remove possibly present electrons from all QDs by visiting configuration I. We then load a single electron at the source QD by going to configuration L. Figure 5c shows jumps in QPC current at different loading configurations (L) that are visited after initialisation via voltage variations from the measurement position, M. The data show that, depending on the voltage variations of the reservoir ($\delta V_R$) and coupling gate ($\delta V_C$), different numbers of electrons can be efficiently loaded into the source QD. Having accomplished the loading process, we go to a sending configuration (S) where the electron can be picked up by the SAW. At the same time as we prepare the source QD for sending, we bring the receiver QD into a configuration allowing the electron to be caught. We then launch a SAW train to execute the transfer of the loaded electron. Comparing QPC currents before and after the SAW burst, we can assess whether the electron was successfully transported.

**Estimation of SAW amplitude**. To estimate the amplitude of potential modulation that is introduced by the SAW, we investigate the broadening of discrete energy levels in QDs by continuous SAW modulation[51]. Owing to the piezoelectric coupling, a SAW passing through a quantum dot leads to a periodic modification of the QDs chemical potential. This causes that the discrete energy states of the quantum dot oscillate with respect to the bias window. During this process—as for the situation of a classical oscillator—the quantum dot states remain most of the time close to turning points of the oscillation. Repeating Coulomb-blockade-peak measurements with increased SAW amplitude, the conductance peaks split according to the amplitude of the periodic potential modulation. The two split lobes indicate the two energies at which a QD state stays on average most of the modulation time. Consequently, one can estimate the peak-to-peak amplitude of

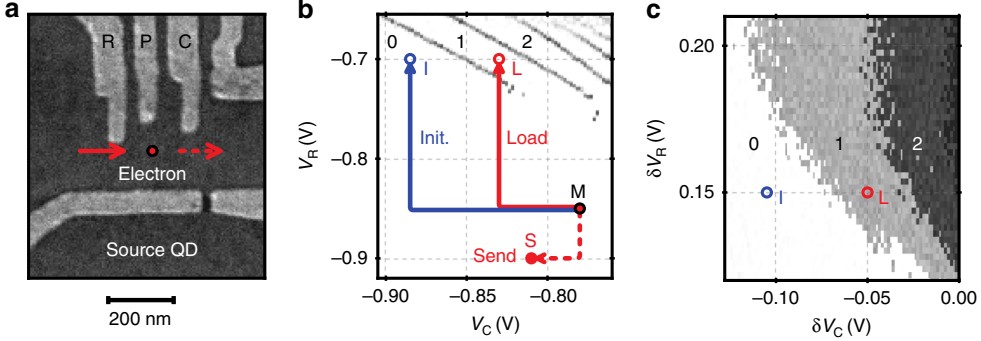

**Fig. 5** Preparation of SAW-driven single-electron transfer. **a** SEM image of a source QD with indication of surface electrodes. **b** Charge-stability diagram showing example source-quantum-dot configurations for QPC measurement (M), initialisation (I), single-electron loading (L) and sending (S). Here, we plot $\partial I_{\mathrm{QPC}}/\partial V_{\mathrm{R}}$. The data show abrupt jumps in QPC current indicating charge-degeneracy lines of the QD. **c** Loading map showing configurations I and L. Each pixel represents the difference in QPC current, $\Delta I_{\mathrm{QPC}}$, before and after visiting the respective loading configuration. The colourscale reflects the electron number in the QD

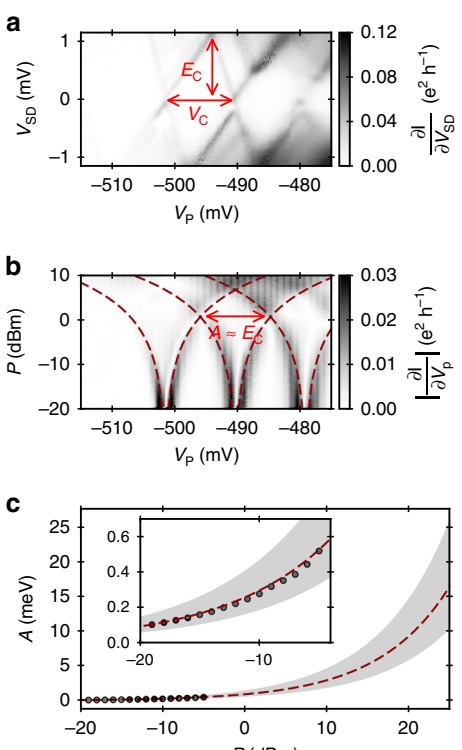

**Fig. 6** Estimation of the SAW amplitude. **a** Exemplary Coulomb-diamond measurement allowing the extraction of the voltage-to-energy conversion factor $\eta = E_{\mathrm{C}}/V_{\mathrm{C}}$. **b** Broadening of the corresponding Coulomb-blockade peaks with increasing transducer power, $P$. **c** Amplitude of SAW-introduced potential modulation. The dashed line shows a fit of Eq. (5) to the experimentally obtained data. The confidence region (grey area) is roughly estimated from variations of measurements on four QDs on a similar sample. The plot shows an extrapolation of this region to the typically employed transducer power of 25 dBm. The inset shows a zoom into the data points

the SAW-introduced potential modulation by determining the energy difference between these two lobes of the split peak.

In order to obtain the peak-to-peak amplitude in energy units, the voltage-to-energy conversion factor $\eta$ has to be known. We determine $\eta$ from Coulomb-diamond measurements as exemplary shown in Fig. 6a. Knowing the voltage-to-energy conversion factor, $\eta$, we can use the SAW-introduced broadening of the Coulomb-blockade peaks to deduce the amplitude of the SAW modulation, $A$, in energy. Figure 6b shows an exemplary data set showing the broadening of Coulomb-blockade peaks with increasing transducer power, $P$. Attenuation along

the transmission line is not taken into account here. The splitting of resonances in Fig. 6b is indicated by the dashed lines. At $P \approx 1$ dBm the side peaks of two neighbouring Coulomb-blockade peaks start to overlap. At the intersection position, the peak-to-peak amplitude of the SAW is equal to the charging energy of the quantum dot, $E_{\mathrm{C}}$. The peak-to-peak amplitude of the SAW-introduced potential modulation, $A$, is related to the transducer power, $P$, by the relation:

$$A\,[\mathrm{eV}] = 2 \cdot \eta \cdot 10^{\frac{P[\mathrm{dBm}] - P_0}{20}},\tag{5}$$

where $P_0$ is a fit parameter accounting for power losses. The voltage-to-energy conversion factor, $\eta = E_{\mathrm{C}}/V_{\mathrm{C}}$, is determined by the aforementioned Coulomb-diamond measurements.

Since these measurements are performed in continuous-wave mode, we trace the broadening of the Coulomb-blockade peaks only up to a transducer power of −5 dBm in order to avoid unnecessary heating. Fitting Eq. (5) via the parameter $P_0$ to the data, we estimate the SAW amplitude for the typically applied transducer power of 25 dBm with 30 ns pulse modulation. Figure 6c shows the SAW amplitude data (zoom in inset) and the extrapolation to 25 dBm (grey area)—the value that was applied in the single-shot-transfer experiments with the present beam-splitter device. The extrapolation indicates a SAW-introduced peak-to-peak modulation of about $(17 \pm 8)$ meV.

**Potential simulations**. Knowing the sample geometry, the electron density in the 2DEG and the set of applied voltages, we calculate the electrostatic potential of the gate-patterned device using the commercial Poisson solver NextNano[43]. We assume a frozen charge layer and deep-boundary conditions[44]. The central premise is that the electron density in the 2DEG is constant, with and without a grounded surface electrode on top of the GaAs/AlGaAs heterostructure. Employing this approach, we deduce a donor concentration of about $1.6 \cdot 10^{10}$ cm$^{-2}$ in the doping layer and a surface charge concentration of about $1.3 \cdot 10^{10}$ cm$^{-2}$. With this information, we can approximately calculate the potential landscape below the surface gates in the experimentally studied voltage configuration. The accuracy of the calculated potential landscape is sufficient to draw qualitative conclusions and to perform an order-of-magnitude discussion.

**Time-dependent simulations**. To simulate the evolution of the SAW-transported electron state, we consider the full two-dimensional potential landscape, $U(\boldsymbol{r},t)$, of our beam-splitter device with a 17 meV peak-to-peak potential modulation of the SAW having a wavelength of 1 μm. We calculate the evolution of the particle described via the electron wave function, $\psi(\boldsymbol{r},t)$, by solving the time-dependent Schrödinger equation:

$$i\hbar \frac{\partial \psi(\boldsymbol{r},t)}{\partial t} = \hat{H}\psi(\boldsymbol{r},t) = \left[ -\frac{\hbar^2}{2m^*}\nabla^2 + U(\boldsymbol{r},t) \right]\psi(\boldsymbol{r},t)\tag{6}$$

where $\hat{H}$ describes the Hamilton operator, $U(\boldsymbol{r},t)$ is the two-dimensional dynamic potential encountered by the electron and $m^*$ is the effective electron mass in a GaAs crystal. We numerically solve the equation using the finite-difference method[45] and discretise the wave function both spatially and in time. In one dimension, the single-particle wave function becomes:

$$\psi(x,t) = \psi(m \cdot \Delta x, n \cdot \Delta t) \equiv \psi_m^n\tag{7}$$

where $m$ and $n$ are integers and $\Delta x$ and $\Delta t$ are the lattice spacing in space and in time, respectively. Following the numerical integration method presented by Askar and Cakmak[52], we evaluate the leading term in the difference between staggered

time-steps:

$$\psi_m^{n+1} = e^{-i\Delta t \hat{H}/\hbar} \psi_m^n \simeq \left(1 - \frac{i\Delta t \hat{H}}{\hbar}\right) \psi_m^n \qquad (8)$$

Consequently, we can write the relation between the time-steps $\psi_m^{n+1}$, $\psi_m^n$, and $\psi_m^{n-1}$ as:

$$\psi_m^{n+1} - \psi_m^{n-1} = \left(e^{-i\Delta t \hat{H}/\hbar} - e^{i\Delta t \hat{H}/\hbar}\right) \psi_m^n \simeq -2\left(\frac{i\Delta t \hat{H}}{\hbar}\right) \psi_m^n \qquad (9)$$

By splitting the wave function in its real and imaginary parts, $\psi_m^n = u_m^n + i v_m^n$, where $u$ and $v$ are real functions, we can evaluate the entire wave function in the same time step. Using the Taylor expansion to estimate the second order spatial derivative, $\frac{\partial^2 \psi}{\partial x^2} \simeq \frac{\psi(x-\Delta x) - 2\psi(x) + \psi(x+\Delta x)}{\Delta x^2}$, the system of equations to solve becomes:

$$u_m^{n+1} = u_m^{n-1} + 2\left(\frac{\hbar \Delta t}{m \Delta x^2} + \frac{\Delta t}{\hbar} U_m^n\right) v_m^n - \frac{\hbar \Delta t}{m \Delta x^2}\left(v_{m-1}^n + v_{m+1}^n\right) \qquad (10a)$$

$$v_m^{n+1} = v_m^{n-1} - 2\left(\frac{\hbar \Delta t}{m \Delta x^2} + \frac{\Delta t}{\hbar} U_m^n\right) u_m^n + \frac{\hbar \Delta t}{m \Delta x^2}\left(u_{m-1}^n + u_{m+1}^n\right) \qquad (10b)$$

By this approach we do not need to obtain the eigenstates of the dynamic QD potential for each time step. Instead, we calculate the eigenbasis only at the beginning of the simulation to form the initial wave function by pure ground state occupation. Solving the system of Eqs. (10a, b) for each successive time step, we then calculate the evolution of the wave function in the dynamic potential landscape that is given by the electrostatic potential defined by the surface gates and the potential modulation of the moving SAW train. We solve the time-dependent Schrödinger equation over the entire tunnel-coupled region using Dirichlet boundary conditions. The boundaries are sufficiently far away from the position of the wave function such that no reflections are observed. To obtain the occupation of the eigenstates after a certain propagation time of the wave-packet, we calculate the eigenstates for the potential of the present time step and decompose the wave function in that basis. The method we use is shown to be convergent and accurate[45].

## Data availability
The data that support the findings of this study are available from the corresponding authors on reasonable request.

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

## Acknowledgements

We would like to acknowledge fruitful discussions with Michihisa Yamamoto. We also acknowledge expert help from Tobias Bautze in the early stage of this project. S.T. acknowledges financial support from the European Union's Horizon 2020 research and innovation programme under the Marie Skłodowska-Curie grant agreement No. 654603 and JSPS KAKENHI Grant Number JP18K14082. A.L. and A.D.W. acknowledge gratefully support of DFG-TRR160, BMBF-Q.Link.X 16KIS0867, LU 2051/1-1 and the DFH/UFA CDFA-05-06. X.W. is funded by the U.S. Office of Naval Research. C.B. and P.S. acknowledge financial support from the French National Agency (ANR) and Deutsche Forschungs Gesellschaft (DFG) in the frame of its programme SingleEIX Project No. ANR-15-CE24-0035. This project has received funding from European UnionâĂŽs Horizon 2020 research and innovation programme under the Marie Skłodowska-Curie grant agreement No. 642688.

## Author contributions

S.T. and H.E. performed the experiment and analysed the data with assistance from P.-A.M., J.W., G.G., and input from P.S., M.Y., M.U., T.M. and C.B. S.T fabricated the sample. H.V.L. developed the theoretical model on time-dependent single-electron SAW propagation under supervision of C.H.W.B. and C.J.B.F. H.V.L., H.E. and X.W. developed the stationary model. A.L. and A.D.W. designed and provided the high-mobility heterostructure. All authors discussed the experimental results. H.E., S.T. and H.V.L. wrote the manuscript with help from G.G. C.B. supervised the experimental work.

## Competing interests

The authors declare no competing interests.
