## [Peer Review File · Nature Communications]

REVIEWERS' COMMENTS:

Reviewer #2 (Remarks to the Author):

The authors have addressed my concerns and improved the quality of the manuscript. They have also added extra simulation results, which show a direction for the future research aiming at the coherent manipulation of a flying electron qubit. The authors' work marks a significant advance in the field of semiconductor physics towards quantum information processing using the flying qubit concept. I recommend publication of this manuscript in Nature Communications.

Reviewer #3 (Remarks to the Author):

In the revised submission the authors have satisfactorily addressed my comments and concerns raised on their original submission. I see this paper as an important contributions in its field and therefore judge the requirements for a publication in Nature Communications fulfilled. Therefore I recommend publication. I have only minor remarks resp. recommendations:

- In line 112/113 they state that a strong dependence of the partitioning process on the potential at the exit of the coupler is found. This statement is very vague, but the point is of interest for others picking up on this. Maybe this could be elaborated? If it needs more space, maybe add as supplemental?
- Figure 2a lacks a scale bar for the vertical (potential) axis. Furthermore, the role of the grey shading (encoding the probability of the states for $\epsilon = 3.5$ meV?) is not mentioned anywhere.
- Figure 2c: The data and labelling of L+U should be darker (much too light in printout). Also the vertical bar of the plus-sign should not align exactly with the grid line (or white out behind the L+U label)
- Line 165: I have doubts on the suitability of the expression "break-up of the two-level system".
- Line 219: "efficiency" is not the right term for the data in 4b, better "transfer probability"
- Fig. 5b: My printer did not print the grey-scale map in 5b (is there on screen), should be checked

We would like to thank the reviewers for careful and critical reading of our revised manuscript. In the following we give a detailed response to all the points raised.

Response to reviewer #2

Reviewer #2 (Remarks to the Author):

The authors have addressed my concerns and improved the quality of the manuscript. They have also added extra simulation results, which show a direction for the future research aiming at the coherent manipulation of a flying electron qubit. The authors' work marks a significant advance in the field of semiconductor physics towards quantum information processing using the flying qubit concept. I recommend publication of this manuscript in Nature Communications.

We are very glad about the positive response!

Response to reviewer #3

Reviewer #3 (Remarks to the Author):

In the revised submission the authors have satisfactorily addressed my comments and concerns raised on their original submission. I see this paper as an important contributions in its field and therefore judge the requirements for a publication in Nature Communications fulfilled. Therefore I recommend publication. I have only minor remarks resp. recommendations:

- In line 112/113 they state that a strong dependence of the partitioning process on the potential at the exit of the coupler is found. This statement is very vague, but the point is of interest for others picking up on this. Maybe this could be elaborated? If it needs more space, maybe add as supplemental?

We agree with the reviewer that the statement is quite vague, since we do not explicitly say how we change the exit potential for this observation. There are multiple ways to do so in experiment. We can for example gradually block electron transfer to one direction by increasing the potential of one of the two exit channels. We therefore added this statement to indicate that the electron partitioning also shows significant sensitivity on the potential around the tunnel-coupled wire.

It is however important to mention that the transition width was not significantly affected by such a change. For the condition of the electron partitioning experiment in our manuscript, we further ensured that the exit potential is symmetric for both directions and made the partitioning process dominated by the potential inside the tunnel-coupled wire.

In retrospect, we find that this statement rather disturbs the focus of the reader and decided to remove it.

- Figure 2a lacks a scale bar for the vertical (potential) axis. Furthermore, the role of the grey shading (encoding the probability of the states for $\epsilon = 3.5$ meV?) is not mentioned anywhere.

Following the comment of the reviewer, we added a scale bar indicating 1 meV in the schematic to clarify the involved energy scales. We added a description of the grey shading of the horizontal lines in the caption:

"... horizontal lines represent the eigenstates in the moving QD, whereas the transparency indicates the suspected exponentially decreasing occupation."

- Figure 2c: The data and labelling of L+U should be darker (much too light in printout). Also the vertical bar of the plus-sign should not align exactly with the grid line (or white out behind the L+U label)

We have changed the colour and the position of the label "L+U" according to the reviewer's proposal.

- Line 165: I have doubts on the suitability of the expression "break-up of the two-level system".

We have reformulated the sentence:

"Simulating the entrance of a flying electron from the injection channel into the tunnel-coupled region, we find significant excitation of the flying electron into higher energy states."

- Line 219: "efficiency" is not the right term for the data in 4b, better "transfer probability"

We have changed the expression according to the reviewer's comment:

"As the voltage pulse overlaps in time with this phase, the sending process is activated and the transfer probability rapidly goes up ..."

- Fig. 5b: My printer did not print the grey-scale map in 5b (is there on screen), should be checked

The problem should now be solved.